# Generating Higher-Fidelity Synthetic Datasets with Privacy Guarantees

**Aleksei Triastcyn * and Boi Faltings**

École Polytechnique Fédérale de Lausanne, 1015 Lausanne, Switzerland; boi.faltings@epfl.ch
* Correspondence: aleksey.tryastsyn@alumni.epfl.ch

**Abstract:** We consider the problem of enhancing user privacy in common data analysis and machine learning development tasks, such as data annotation and inspection, by substituting the real data with samples from a generative adversarial network. We propose employing Bayesian differential privacy as the means to achieve a rigorous theoretical guarantee while providing a better privacy-utility trade-off. We demonstrate experimentally that our approach produces higher-fidelity samples compared to prior work, allowing to (1) detect more subtle data errors and biases, and (2) reduce the need for real data labelling by achieving high accuracy when training directly on artificial samples.

**Keywords:** machine learning; synthetic data; generative adversarial networks; privacy

## 1. Introduction

With machine learning (ML) becoming ubiquitous in many aspects of our society, questions of its privacy and security take centre stage. A growing field of research in privacy attacks on ML [1–4] tells us that it is possible to infer information about training data even in a black-box setting, without access to model parameters. A wider population, however, is concerned with privacy practices used in the ML development cycle, such as company employees or contractors manually inspecting and annotating user data (https://www.theguardian.com/technology/2020/jan/10/skype-audio-graded-by-workers-in-china-with-no-security-measures, accessed on 29 May 2022, https://www.bloomberg.com/news/articles/2019-04-10/is-anyone-listening-to-you-on-alexa-a-global-team-reviews-audio, accessed on 29 May 2022).

The problem of privacy attacks is often tackled by adding a differentially private mechanism to the model training procedure [5]. Differential privacy (DP) [6] provides a rigorous theoretical guarantee, which states (informally) that the algorithm output would not significantly change when a single user adds or removes their data, except with small (failure) probability. Another approach gaining popularity is *federated learning* (FL), introduced by McMahan et al. [7], where a central entity trains a model by computing updates locally on-device and then securely aggregating these updates on a server. This way user data never leave their devices.

In spite of significant progress, neither of these approaches solves the problem of manual data labelling. Moreover, it creates an additional hurdle for developers, as they cannot inspect data, especially in decentralised settings, making it difficult to understand the model behaviour and find bugs in data and implementations. Augenstein et al. [8] provide a more complete characterisation of these questions.

This paper follows [8] in adopting generative adversarial networks (GAN) [9] trained in a privacy-preserving manner for addressing these issues. More specifically, we use the notion of Bayesian differential privacy (BDP) [10], which takes into account the data distribution and provides a more meaningful guarantee for in-distribution samples than classical DP. Intuitively, the traditional DP has uniform failure probability for all data points, including the ones that come from a completely different distribution and even pure noise.

This kind of data creates difficulties for machine learning as they will significantly reduce the accuracy of the resulting model, and other measures must be taken before applying a privacy mechanism so that it does not enter the dataset. BDP allows us to focus the privacy guarantee on datasets that do not contain instances that are "impossible" or come from a different distribution, and thus provides a more realistic privacy bound. The fact that the privacy guarantee may not hold for such outliers can be seen as an advantage as it would allow us to recognise and filter them to eliminate their influence on the learned model. Since both DP and BDP can use the same obfuscation mechanism, while computing two privacy bounds in parallel, a DP guarantee would still hold for out-of-distribution samples. More details on the overall approach and privacy are provided in Section 2.3.

The advantage of using this privacy definition is that it enables generating data of higher fidelity, compared to previous work on GANs with DP, allowing for finer-grained inspection of data. While some problems with data or data pipelines can be discovered using very coarse samples (e.g., pixel intensity inversion in [8]), more subtle bugs, such as partial data corruption, would require samples of much better quality, rendering the DP guarantee too loose to be meaningful. Moreover, if fidelity is high enough, synthetic data can be used for annotation and training itself, removing the related privacy concerns and extending applicability of FL. We evaluate our solution in these two aspects in Section 3.

The main contributions of this paper are as follows:

- We use Bayesian DP to enable higher quality GAN samples, while still providing a strong privacy guarantee;
- We demonstrate that this technique can be used to discover finer data errors than has been previously reported;
- We also show that for some tasks synthetic data are of high enough quality to be used for labelling and training.

## 2. Materials and Methods

In this section, we describe our main method and the employed privacy analysis. First, however, let us start with a brief overview of the relevant prior work and the background necessary for understanding the remainder of the article.

### 2.1. Related Work

A rapidly expanding area of privacy-preserving machine learning research has been recently focused on the attacks that compromise privacy of training data, such as model inversion [1] and membership inference [2]. The former is based on observing the output probabilities of the target model for a given class and performing gradient descent on an input reconstruction. The latter assumes an attacker with access to similar data, which is used to train "shadow" models, mimicking the target, and the attack model, which predicts if a certain example has already been seen during training based on its output probabilities. Both attacks can be performed in a black-box setting, without access to the model internal parameters.

Differential privacy (DP) [6] is widely accepted as the gold standard for preventing such attacks. One of the early takes on the problem is to use disjoint datasets and distributed training with DP. For example, ref. [11] proposes to train a model in a distributed manner by communicating sanitised updates from participants to a central authority. Such a method, however, yields high privacy losses [5,12]. An alternative technique suggested by [12] also uses disjoint training sets and builds an ensemble of independently trained teacher models to transfer knowledge to a student model by labelling public data. This result has been extended in [13] to achieve state-of-the-art image classification results in a private setting (with single-digit DP bounds). A different approach is taken by [5]. They propose using differentially private stochastic gradient descent (DP-SGD) to train deep learning models in a private manner. This approach achieves high accuracy maintaining relatively low DP bounds and being simpler to implement, but may also require pre-training on public data.

Due to the fact that the DP threat model is extremely broad, achieving a reasonable guarantee may be difficult or even impossible. For this reason, a number of alternative definitions has been proposed over recent years, aimed at relaxing the guarantee or providing tighter composition bounds under certain assumptions. Examples are computational DP [14], mutual-information privacy [15,16], different versions of concentrated DP (CDP [17], zCDP [18], tCDP [19]), and Rényi DP (RDP) [20]. Some other relaxations [21–23] tip the balance even further in favour of applicability at the cost of weaker guarantees, for example considering the average-case instead of the worst-case [24].

In this work, we rely on another relaxation, called Bayesian differential privacy [10]. This notion utilises the fact that data come from a particular distribution, and not all data samples are equally likely (e.g., unlikely to find a sound record among ECG samples). At the same time, it maintains a similar probabilistic interpretation of its parameters $\varepsilon$ and $\delta$. It is worth noting, that unlike some of the relaxations mentioned above, Bayesian DP can provide a tail bound on privacy loss, similarly to the moments accountant (MA) [5], and is not limited to a particular dataset, but rather a particular type of data (e.g., emails, MRI images, etc.), which is a much more permitting assumption.

Up until recently, another aspect of privacy in machine learning has been largely overlooked: the human involvement in the development cycle and manual data processing. These issues can be mitigated, at least partially, by federated learning (FL) [7], which brings a great promise for user privacy. Yet, FL paradigm creates additional problems of its own. Augenstein et al. [8] provide a good starting point, systematising these problems and proposing a solution by the use of synthetic data. Although privacy-preserving data synthesis using GANs has been introduced in earlier works [24–29], these papers mainly focused on achieving high utility of synthetic data without addressing a broader scope of privacy leakage via manual data handling.

A common problem of privacy-preserving GANs, however, is that the generated samples have very low fidelity, unless the privacy guarantee is unreasonably weak. Our approach makes progress in exactly this perspective: we can achieve much higher quality outputs with little compromise in privacy guarantees (and only for outliers that are difficult to hide). As a result, our synthetic data yield better performance of downstream analytics, and simultaneously, provide more powerful data inspection capabilities.

*2.2. Background*

We use $D, D'$ to represent neighbouring (adjacent) datasets. If not specified, it is assumed that these datasets differ in a single example. Individual examples in a dataset are denoted by $x$ or $x_i$, while the example by which two datasets differ—by $x'$. We assume $D' = D \cup \{x'\}$, whenever possible to do so without loss of generality. The private learning outcomes (i.e., noised gradients) are denoted by $w$.

**Definition 1.** *A randomised function (mechanism) $\mathcal{A} : \mathcal{D} \to \mathcal{R}$ with domain $\mathcal{D}$ and range $\mathcal{R}$ satisfies $(\varepsilon, \delta)$-differential privacy if for any two adjacent inputs $D, D' \in \mathcal{D}$ and for any set of outcomes $\mathcal{S} \subset \mathcal{R}$ the following holds:*

$$\Pr[\mathcal{A}(D) \in \mathcal{S}] \le e^{\varepsilon} \Pr[\mathcal{A}(D') \in \mathcal{S}] + \delta. \tag{1}$$

**Definition 2.** *Privacy loss of a randomised mechanism $\mathcal{A} : \mathcal{D} \to \mathcal{R}$ for inputs $D, D' \in \mathcal{D}$ and outcome $w \in \mathcal{R}$ takes the following form:*

$$L(w, D, D') = \log \frac{\Pr[\mathcal{A}(D) = w]}{\Pr[\mathcal{A}(D') = w]}. \tag{2}$$

**Definition 3.** *The Gaussian noise mechanism achieving $(\varepsilon, \delta)$-DP, for a function $f : \mathcal{D} \to \mathbb{R}^m$, is defined as*

$$\mathcal{A}(D) = f(D) + \mathcal{N}(0, I\sigma^2), \tag{3}$$

*where $\sigma > C\sqrt{2\log\frac{1.25}{\delta}}/\varepsilon$ and C is the L2-sensitivity of f.*

For more details on differential privacy, the Gaussian mechanism, and how to use it in machine learning, we refer the reader to [5,30].

Let us now present the notion of Bayesian DP, initially introduced in [10].

**Definition 4.** *A randomised function (algorithm) $\mathcal{A} : \mathcal{D} \to \mathcal{R}$ with domain $\mathcal{D}$ and range $\mathcal{R}$ satisfies $(\varepsilon_\mu, \delta_\mu)$-Bayesian differential privacy if for any two adjacent datasets $D, D' \in \mathcal{D}$, differing in a single data point $x' \sim \mu(x)$, and for any set of outcomes $\mathcal{S} \subset \mathcal{R}$ the following holds:*

$$\Pr[\mathcal{A}(D) \in \mathcal{S}] \leq e^{\varepsilon_\mu}\Pr[\mathcal{A}(D') \in \mathcal{S}] + \delta_\mu. \tag{4}$$

Subscript $\mu$ highlights the main difference with the classic DP: in the original definition the probability is taken only over the randomness of the outcome ($w$), while in BDP over two random variables ($w$ and $x'$). Therefore, the privacy parameters $\varepsilon$ and $\delta$ depend on the data distribution $\mu(x)$, although it is not required to be known. This modification changes the meaning of $\delta_\mu$ compared to the $\delta$ of DP: it now also accounts for the privacy mechanism failures in the tails of data distributions in addition to the tails of the outcome distributions. Lastly, data samples are assumed to be exchangeable [31], a requirement for tighter privacy accounting, which is satisfied in many real-world scenarios.

Basic properties of DP are maintained by BDP, making it widely applicable:

**Proposition 1** (Post-processing). *Let $\mathcal{A} : \mathcal{D} \to \mathcal{R}$ be a $(\varepsilon_\mu, \delta_\mu)$-Bayesian differentially private algorithm. Then for any arbitrary randomised data-independent mapping $f : \mathcal{R} \to \mathcal{R}'$, $f(\mathcal{A}(D))$ is $(\varepsilon_\mu, \delta_\mu)$-Bayesian differentially private.*

**Proposition 2** (Basic composition). *Let $\mathcal{A}_i : \mathcal{D} \to \mathcal{R}_i$, $\forall i = 1\ldots k$, be a sequence of $(\varepsilon_\mu, \delta_\mu)$-Bayesian differentially private algorithms. Then their combination, defined as $\mathcal{A}_{1:k} : \mathcal{D} \to \mathcal{R}_1 \times \ldots \times \mathcal{R}_k$, is $(k\varepsilon_\mu, k\delta_\mu)$-Bayesian differentially private.*

**Proposition 3** (Group privacy). *Let $\mathcal{A} : \mathcal{D} \to \mathcal{R}$ be a $(\varepsilon_\mu, \delta_\mu)$-Bayesian differentially private algorithm. Then for all pairs of datasets $D, D' \in \mathcal{D}$, differing in k data points $x_1, \ldots, x_k$ s.t. $x_i \sim \mu(x)$ for $i = 1\ldots k$, $\mathcal{A}(D)$ is $(k\varepsilon_\mu, ke^{k\varepsilon_\mu}\delta_\mu)$-Bayesian differentially private.*

See Section 2.4 for details on advanced composition and privacy accounting.

Finally, in parts of the paper, we refer to the classification of ML developer tasks presented in [8], which can be condensed to:

T1—sanity checking data;

T2—debugging mistakes;

T3—debugging unknown labels/classes;

T4—debugging poor performance on certain classes/slices/users;

T5—human labelling of examples;

T6—detecting bias in the training data.

*2.3. Our Approach*

We propose to employ Bayesian differential privacy instead of the traditional DP. Its primary distinction is that it takes into account the data distribution, and by extension, assumes that all data points are drawn from the same distribution, although these distributions may be multimodal, highly complex, and generally unknown. This is a natural hypothesis in many machine learning applications, but especially so when working with generative models such as GANs.

The task of generative modelling in itself is to learn an underlying data distribution, and thus, a common distribution is an implicit belief. This results in an organic match with BDP, because there are no assumptions to add to the problem.

Another part of our intuition is that the foremost source of privacy leakage are outliers. On the one hand, their respective privacy loss would be discounted in BDP accounting due to their low probability. On the other hand, we can reduce the number of samples generated by the GAN to decrease the chances of these outliers appearing in the synthetic dataset.

### 2.3.1. Formal Overview

We are given a dataset $D$ of labelled ($\{(x_i, y_i)|(x_i, y_i) \sim \mu(x,y),\ i = 1...n\}$) or unlabelled ($\{x_i \mid x_i \sim \mu(x),\ i = 1...n\}$) examples. This dataset can be decentralised, in which case we would use FL (see the next subsection). Our task is to train a GAN, which consists of the generator $\mathcal{G}$ and the critic $\mathcal{C}$ (discriminator), to generate synthetic samples from $\mu$.

Our privacy mechanism follows the previous work on privacy-preserving GANs [25,26]. More specifically, it applies the Gaussian mechanism (clip to norm $C$ and add Gaussian noise with variance $C^2\sigma^2$) to discriminator updates at each step of the training. Privacy of the generator is then guaranteed by the post-processing property of BDP. It is worth mentioning, however, that clipping and/or adding noise to generator gradients should not be overlooked by developers either. It can be beneficial for training in some cases, as it helps keeping a better balance in the game between the critic and the generator.

We choose not to implement more complicated schemes, such as PATE-GAN [28] or G-PATE [29], which use the PATE framework [13] to guarantee differential privacy for GANs. Our key rationale is that a more complicated structure of this solution could create unnecessary errors and additional privacy leakage (e.g., leaking privacy by back-propagating through the teachers' votes to the generator, thereby neglecting the added noise). Nevertheless, we show in our evaluation that due to the distribution-calibrated BDP accounting (and hence, less added noise) our GAN generates better quality samples compared to these more complex solutions.

### 2.3.2. Federated Learning Case

It is worth noting that we did not make any assumptions on where the data are located. The most logical scenario to consider is federated learning, such as in [8], such that the data remain on user devices at all times.

To accommodate FL scenarios, minimal modifications to the approach are required. Training of the generative model would be performed in the same way as any other federated model, and privacy accounting would be performed at the user-level [8]. Bayesian DP results (see the next section) are also directly transferable to FL [32], and privacy bounds are generally even tighter in this case.

### 2.4. Privacy Analysis

In order to compute privacy guarantees of the synthetic dataset w.r.t. the real one, we need to bound privacy loss of the generative model. As noted before, we effectively enforce privacy on the critic and then rely on preservation of guarantees under post-processing. This arrangement ensures a simple adoption of privacy accounting for discriminative models.

Privacy accounting is performed by using the Bayesian accountant [10]. To benefit from the data distribution information, it needs to sample a number of gradients at each iteration in addition to the one used in the update. These gradients are then used to estimate the upper confidence bound on the privacy cost, generally defined as $c_r(\lambda) = \log \mathbb{E}_{x'} \left[ e^{r\lambda \mathcal{D}_{\lambda+1}(p_D \| p_{D'})} \right]^{\frac{1}{r}}$ for some exponent $r$.

More precisely, for the Gaussian noise mechanism that we use in this paper, let

$$c_t(\lambda) = \max\{c_t^L(\lambda), c_t^R(\lambda)\}, \tag{5}$$

where

$$c_t^L(\lambda) = \frac{1}{T} \log \mathbb{E}_{x'} \left[ \mathbb{E}_{k \sim B(\lambda+1,q)} \left[ e^{\frac{k^2-k}{2\sigma^2} \|g_t - g_t'\|^2} \right]^T \right], \tag{6}$$

$$c_t^R(\lambda) = \frac{1}{T} \log \mathbb{E}_{x'} \left[ \mathbb{E}_{k \sim B(\lambda,q)} \left[ e^{\frac{k^2+k}{2\sigma^2} \|g_t - g_t'\|^2} \right]^T \right]. \tag{7}$$

Here, $B(\lambda, q)$ is the binomial distribution with $\lambda$ experiments (a hyper-parameter) and the probability of success $q$ (equal to the probability of sampling a single data point in a batch), $g_t$ and $g_t'$ are two gradient samples differing in one data point, and $T$ is the total number of learning iterations/rounds. These expressions can be derived from the definition, as shown in [10,33].

In practice, gradient sampling can be performed in the following way. At each optimisation step, we simply add or remove $m$ random data points to/from the batch and recompute the corresponding gradient. Let $\overline{G}_t$ denote a sample mean of the expressions inside $\mathbb{E}_{x'}[\cdot]$ in Equations (6) and (7), computed with these gradients, and $S_t$ be a sample standard deviation. We can then estimate the privacy costs (6) and (7) for the given gradient distribution using a Bayesian estimator:

$$\hat{c}_t(\lambda; \gamma, m) = \frac{1}{T} \log \left[ \overline{G}_t + \frac{F^{-1}(1-\gamma, m-1)}{\sqrt{m-1}} S_t \right], \tag{8}$$

where $F^{-1}(1 - \gamma, m - 1)$ is the inverse of the Student's $t$-distribution CDF at $1 - \gamma$ with $m - 1$ degrees of freedom. This estimate could be interpreted as an upper bound on the privacy cost with probability $(1 - \gamma)$ under the assumption of an uninformative prior. Again, we refer the reader to [10,33] or the accountant implementation (https://github.com/AlekseiTriastcyn/bayesian-differential-privacy, accessed on 29 May 2022) for further details.

Using these per-iteration privacy costs, we can calculate the total privacy cost via the advanced composition theorem:

**Theorem 1** (Advanced Composition). *Let $\mathcal{A}^{(1:T)} = (\mathcal{A}^{(1)}, \dots, \mathcal{A}^{(T)})$ be a sequence of privacy mechanisms. Then the total privacy cost $c(\lambda)$ of $\mathcal{A}^{(1:T)}$ satisfies*

$$c(\lambda) \le \sum_{t=1}^{T} c_t(\lambda).$$

The privacy guarantee is calculated from the privacy cost, by fixing either $\varepsilon_\mu$ or $\delta_\mu$:

$$\log \delta_\mu \le \sum_{t=1}^{T} c_t(\lambda) - \lambda \varepsilon_\mu. \tag{9}$$

Note that here we use a more lightweight version of notation for readability. In practice, one needs to keep in mind, for example, possible auxiliary information and maximise privacy cost over it. For more in-depth analysis and related proofs, see [33].

An important difference in privacy accounting for GANs is that not every update of the critic should be accounted for. Updates on fake data samples do not leak information about the real data beyond what is already accounted for in the previous iterations. Therefore, only real updates are sampled and used for the privacy cost estimation. In some GAN architectures, however, one should be careful to consider additional sources of privacy leakage, such as the gradient penalty in WGAN-GP [34].

To better understand how the BDP bound relates to the traditional DP, consider the following conditional probability:

$$\Delta(\varepsilon, x') = \Pr\big[L(w, D, D') > \varepsilon \mid D, D' = D \cup \{x'\}\big]. \tag{10}$$

The moments accountant outputs $\delta$ that upper-bounds $\Delta(\varepsilon, x')$ for all $x'$. It is not true in general for other accounting methods, but let us focus on MA, as it is by far the most popular. Consequently, the moments accountant bound is

$$\max_{x'} \Delta(\varepsilon, x') \leq \delta, \tag{11}$$

where $\varepsilon$ is a chosen constant. At the same time, BDP bounds the probability that is not conditioned on $x'$, but we can transform one to another through marginalisation and obtain:

$$\mathbb{E}_{x'}\big[\Delta(\varepsilon, x')\big] \leq \delta_\mu. \tag{12}$$

On the surface, this guarantee seems considerably weaker, as it holds only in expectation. However, since $\Delta(\cdot)$ is a non-negative random variable in $x$, we can apply Markov's inequality and obtain a tail bound on it using $\delta_\mu$. *We can therefore find a pair $(\varepsilon, \delta)_p$ that holds for any percentile p of the data/user distribution, not just in expectation.* In all our experiments, we consider bounds well above the 99th percentile, so it is very unlikely to encounter data for which the equivalent DP guarantee does not hold.

### 2.5. Experimental Procedures

We evaluate two major applications of the technique. First, we show that the generated samples can be used for debugging the ML model through data inspection, resembling tasks T1–T4 from [8]. Second, we examine the quality of the downstream ML model trained directly on synthetic samples, thus demonstrating a possibility of solving T5 (data labelling/annotation) as well.

In the debugging experiment, we attempt to detect a more subtle bug compared to [8]: an incorrect image rotation that yields lower model performance. While the pixel intensity inversion can be easily spotted using low-fidelity synthetic samples, image rotation requires higher fidelity to be detected.

Downstream learning experiments are set up as follows:

1. Train the generative model (*teacher*) on the original data under privacy guarantees.
2. Generate an artificial dataset by the obtained model and use it to train ML models (*students*).
3. Evaluate students on the held-out real test set.

We use two image datasets, MNIST [35] and Fashion-MNIST [36]. Both have $60,000$ training and $10,000$ test examples, where each example is a $28 \times 28$ size greyscale image. The task of MNIST is handwritten digit recognition, while for Fashion-MNIST it is clothes type recognition. Although these datasets may not be of particular interest from the privacy viewpoint, this choice is defined by the ability to compare to prior work.

Our evaluation is implemented in Python and Pytorch (http://pytorch.org, accessed on 29 May 2022). Source code is included in Supplementary Material; it will be released publicly upon publication. For the generative model, we experimented with variations of Wasserstein GAN [37] and WGAN-GP [34], but found the former to produce better results, probably because gradient clipping is already a part of the privacy mechanism. Our critic consists of three convolutional layers with SELU activations [38] followed by a fully connected linear layer with another SELU and then a linear classifier. The generator starts with a fully connected linear layer that transforms noise (and possibly labels) into a 4096-dimensional feature vector which is then passed through a SELU activation and three deconvolution layers with SELU activations. The output of the third deconvolution layer is down-sampled by max pooling and normalised with a `tanh` activation function.

All models are trained using Adam with the learning rate 0.0001. The clipping threshold for gradients of discriminators is set to 0.5, no clipping is performed for generators,

and the noise standard deviation is 0.01 for MNIST and 0.02 for Fashion-MNIST. The reported accuracy and privacy were achieved with 400 epochs for both datasets.

Although we use centralised setting throughout this section, the results are readily transferable to federated scenarios. Previous work suggests that neither GAN sample quality [24] nor BDP guarantees [32] should be significantly affected.

## 3. Results

### 3.1. Data Inspection

The data inspection experiment is setup in the following way. We introduce the rotation bug through randomly rotating some images by 90°. We then train the two generative models, on correct images and on altered images, and compare their samples. We also train a model with DP to show that its image quality would not be sufficient to detect the error.

Figure 1 shows the output of generative models trained on MNIST with and without image rotation. By examining the samples, developers can clearly determine that a portion of images was rotated. This way, the error can be promptly identified and fixed. On the other hand, with generative models that uphold the traditional DP guarantee (Figure 1c), it would be difficult to detect such pre-processing error, because the produced samples have very low fidelity, even though $\varepsilon$ in this case is unjustifiably high at the order of $10^7$.

We also observe that the synthetic data quality under BDP (see Figures 1a and 2) is sufficient to detect previously unseen classes or dataset biases, such as under-represented classes. Moreover, these results are achieved with a strong privacy guarantee: under $(1, 10^{-10})$-BDP, and hence, the probability that $(1, 10^{-5})$-DP does not hold for this data is less than $10^{-5}$.

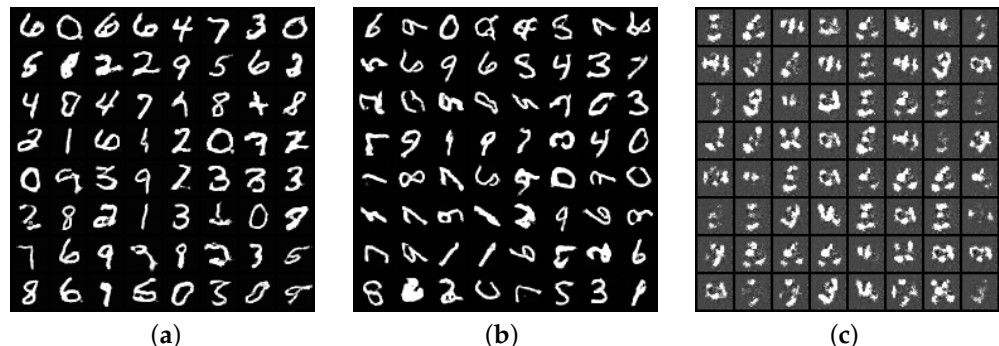

(a)        (b)        (c)

**Figure 1.** GAN output for detecting unwanted rotations on MNIST. (**a**) Trained on correct images with BDP; (**b**) Trained on altered images with BDP; (**c**) Trained on altered images with DP.

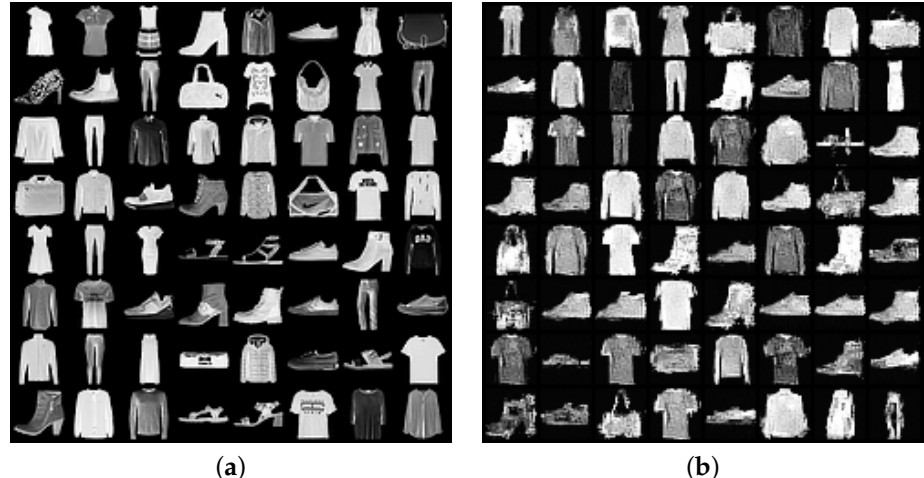

(a)        (b)

**Figure 2.** Real and synthetic samples on Fashion-MNIST. (**a**) Real; (**b**) Synthetic.

### 3.2. Learning Performance

Here, we evaluate the generalisation ability of the student model trained on artificial data. More specifically, we train a student model on generated data and report test classification accuracy on a real held-out set.

The goal of this experiment is to show that having a privacy-preserving generative model we can use synthetic samples to fully replace the real data. Not only does it allow us to eliminate manual labelling of real (and potentially sensitive) data, but also expand the set of problems that can be solved by FL (task T5 in Augenstein et al. [8] classification). For example, some medical data cannot be automatically annotated, and users are not qualified to do that, so high-quality synthetic data would allow the annotation to be performed by doctors without privacy risks for users.

We imitate human annotation by training a separate classifier (with the same privacy guarantee as the generative model) and using it to label synthetic images. While this approach is somewhat different from prior work on generating data for training ML models, comparisons in this section are still valid because our annotator maintains the same privacy guarantee.

We choose to compare with the method called G-PATE [29], because it is one of the best recent techniques in terms of privacy-utility trade-off. The authors showed that it outperforms another PATE-based approach, PATE-GAN [28], as well as DP-GAN [26], based on DP-SGD.

Student model accuracy is shown in Table 1. Apart from G-PATE, we compare our method to a non-private classifier trained directly on the real dataset, and a private classifier, trained on the real dataset with Bayesian DP. In the case of generative models, the same (non-private) classifier is trained on the private synthetic output. All results in the table are obtained with the privacy guarantee of $(1, 10^{-5})$-DP, or $(1, 10^{-10})$-BDP, which is equivalent to $(1, 10^{-5})$-DP for this data with high probability. Although Long et al. [29] report improved results for $(10, 10^{-5})$-DP, we do not include those in the study for two reasons. First, $\varepsilon = 10$ is too high for providing a meaningful guarantee [10]. Second, their accuracy is still lower than the one our method attains under stricter $\varepsilon = 1$.

**Table 1.** Accuracy of models: (1) non-private baseline (convolutional network); (2) private classifier (convolutional network trained with BDP); and student models: (3) for G-PATE with $(1, 10^{-5})$-DP guarantee; (4) for WGAN with $(1, 10^{-10})$-BDP guarantee (our method).

| Dataset | Non-Private | Private Classifier | G-PATE | Our Approach |
|---|---|---|---|---|
| MNIST | 99.20% | 95.59% | 56.31% | 93.64% |
| Fashion-MNIST | 91.51% | 82.20% | 51.74% | 76.83% |

Generally, we observe that on these datasets switching from real to synthetic data does not significantly deteriorate accuracy of the student model while maintaining strong theoretical privacy guarantees. On MNIST, the drop in performance between a private discriminative and a private generative approach is less than 2%. It is more noticeable on Fashion-MNIST, but is still within about 7% and is still lower than the drop between non-private and private classifiers.

In another experiment, Figure 3a,b show a percentage of maximum accuracy (i.e., numbers reported in Table 1) achievable by the model when only a part of the generated data is labelled. Notably, the models trained on synthetic data achieve the same percentage faster than the ones trained on the real data, which is especially evident for the Fashion-MNIST dataset. Moreover, as little as 100 labelled samples is enough to outperform models trained on data generated with comparable DP guarantees.

Using non-private synthetic data (not shown in the table) allows us to achieve somewhat better results. On MNIST, the student model reaches 96.09%, while on Fashion-MNIST, 84.86%. It suggests that about half of the accuracy loss comes from the limited capacity of

the generative model. Figure 3a,b appear to corroborate this finding, as the learning curve for synthetic data saturates quicker (i.e., a generative model is sufficiently described by a smaller number of samples than a real data distribution).

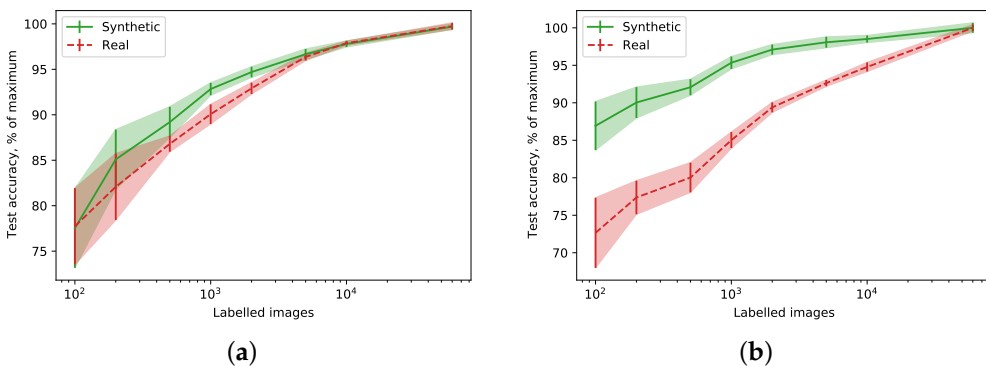

(**a**)                                                        (**b**)

**Figure 3.** Relative accuracy (a percentage of maximum achievable accuracy) for different numbers of labelled images. (**a**) MNIST; (**b**) Fashion-MNIST.

## 4. Discussion and Conclusions

We explore the use of generative adversarial networks to tackle the problem of privacy-preserving data inspection and annotation in machine learning. While the previous approaches to this problem involve generative models either without any privacy guarantee or with differential privacy, we opt for a different privacy notion—Bayesian differential privacy. By capturing the inherent properties of data and allowing for non-uniform privacy loss throughout the dataset, it enables higher-fidelity synthetic data while still maintaining a privacy guarantee comparable to DP.

Our evaluation shows that privacy-preserving GANs with BDP can be used to detect subtle bugs in data itself or pre-processing pipelines, which could not be caught by DP GANs due to low sample fidelity. Similarly, biases in the data and previously unseen classes can be discovered.

In addition, the generated data can be directly annotated and used for training in place of the real data. We demonstrate that student models trained on our synthetic samples achieve significantly higher accuracy compared to prior state-of-the-art and exhibit only a mild drop in performance compared to private classification with real data. Furthermore, this gap is mainly determined by the quality of the generative model, and hence, will get smaller with advances in that field.

The application of our method can be slightly more involved than a standard DP mechanism with Rényi accountant, as it requires sampling additional data points for estimating privacy costs. However, implementation examples are available to aid researchers. At the same time, practitioners can benefit from the fact that our privacy mechanism itself is exactly the same as for conventional DP, differing only in accounting, and is thus substantially simpler than the ones derived from PATE. Therefore, we could recommend to use this method when computational resources allow for additional gradient computations and a non-uniform privacy guarantee favouring more typical data is desired.

There are several interesting future directions for this research. First, different generative models and different datasets should be evaluated. We used a rather basic Wasserstein GAN, but there are more advanced GANs for image data, GANs for discrete inputs, as well as other classes of generative methods. With more elaborate models, one could tackle datasets such as CIFAR or even ImageNet. Another important question to investigate is automation of the privacy parameters choice, such as noise variance and gradient clipping threshold. We treated them similarly to learning hyper-parameters, but it complicates the training process and increases the privacy budget. Thus, a less manual, perhaps adaptive, procedure for choosing these parameters will provide a significant contribution; not only for privacy-preserving data release, but for privacy-preserving ML in general.

By and large, research in high-quality generative methods with privacy guarantees can potentially have a very broad impact, accelerating the private federated learning development on mobile, wearable, and IoT devices, and facilitating medical and research data sharing for increased efficiency, transparency, and reproducibility.

**Supplementary Materials:** The following supporting information can be downloaded at: https://www.mdpi.com/article/10.3390/a15070232/s1.

**Author Contributions:** Conceptualization, A.T.; Methodology, A.T.; Software, A.T.; Validation, A.T.; Formal Analysis, A.T.; Investigation, A.T.; Writing—Original Draft, A.T.; Writing—Review & Editing, A.T. and B.F.; Supervision, B.F.; Project Administration, A.T. and B.F.; Funding Acquisition, B.F. All authors have read and agreed to the published version of the manuscript.

**Funding:** This research received no external funding.

**Data Availability Statement:** Not applicable.

**Conflicts of Interest:** The authors declare no conflict of interest.

## Abbreviations

The following abbreviations are used in this manuscript:

DP      Differential privacy
BDP     Bayesian differential privacy
FL      Federated learning
ML      Machine learning
GAN     Generative adversarial network
SGD     Stochastic gradient descent

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
