# Peer review of "Generating Higher-Fidelity Synthetic Datasets with Privacy Guarantees"

_algorithms, doi:10.3390/a15070232_

Round 1

Reviewer 1 Report

This paper addresses data synthesis with a differential privacy guarantee, a trending and essential topic in data privacy. The proposed approach utilizes a weak privacy notion, Baysisan DP, to improve data quality of GAN. 

Although the paper mainly applies BDP and Bayesian accountant for GAN training, it should be self-contained and have BDP accountant clearly explained. The paper mentioned that it needs to sample a number of gradients at each iteration to estimate the upper bound on the privacy cost. It is not clear to me how the gradients are sampled, they are protected, and how they affect the privacy cost.

The evaluation is limited. The authors did not compare the model accuracy under different privacy costs under BDP. It would have been better if the experiment had shown the effects of varying privacy costs.

The statement non-private synthetic data (not shown in the table) allow to reach only marginally better results, suggesting that most of the accuracy loss comes from the generative model rather than privacy preservation. It is difficult to follow the statement without showing any numbers or detailed explanations. 

There are some minor typos, for example: 

Page 2: "data errors than has been" => that

Page 3: "a number of alternative definitions has" => have

Author Response

We are grateful for the insightful comments provided by the reviewer. In order to address them we made the following revisions:
 - Sections 2.2 and 2.4 were extended with more detailed descriptions of BDP and privacy accounting, including an additional GitHub reference for the accountant implementation.
 - The statement regarding non-private synthetic data has been revised and supported by numbers from our experiments.
Unfortunately, we don’t have the results showing the effects of varying privacy cost as this was not the goal of our study. However, we believe GANs would behave similarly to feed-forward models in this aspect and we have studied that in earlier work [10].
Finally, the mentioned typos are not actually typos, these sentences are in fact correct. But we will go over the manuscript again to check for mistakes.

Reviewer 2 Report

Summary:

This paper proposes to use Bayesian Differential Privacy (BDP) to produce higher quality samples generated by privacy-preserved Generative Adversarial Networks (GAN). It allows finer-grained inspection of data. The high-quality synthetic data can be further used for labeling and training. The authors provided experimental evaluations on MNIST and F-MNIST to support these claims.

Strength:

  1. It is interesting and intuitive that the BDP can be applied to generate synthetic data with higher fidelity. Examples in Figure 2 clearly show the improvement.
  2. The paper is well-written, easy to read and follow.

Weakness:

  1. One minor concern is that MNIST and F-MNIST are relatively simple datasets. To demonstrate the practical usefulness of the proposed method, it would be better to have evaluations on larger and more difficult ones like CIFAR-10 or even ImageNet.

Author Response

We thank the reviewer for their time and comments on our manuscript. We are happy that the reviewer found our method to be interesting and intuitive, and our paper easy to read. We also understand the concern related to the datasets chosen for the study and recognise the need for extended experiments on more complex data. Unfortunately, GAN architectures we considered don’t have sufficient expressive power for CIFAR or ImageNet datasets, and the time allotted for minor revisions would not allow us to evaluate bigger models. That said, we will keep this in mind as future work and note that in the manuscript.

Reviewer 3 Report

The paper discusses  one very important issue in the area of data privacy and machine learning modelling and proposes an approach for enhancement the person' s privacy. The problem is explained in details relaying on contemporary scientific sources. The presented solution is original, giving promising results.

It is nice, if the section 4 is titled: Discussion and conclusions. It will be very useful for researchers, if the authors form some recommendations about the application of their approach in comparison to others.

Author Response

We appreciate the reviewer's effort in assessing our paper, and are glad to see that they consider our solution original and promising. We took the reviewers comments into account by renaming Section 4 and adding a paragraph (in the same section) with our recommendations regarding application of the method.